# From Social Rejection to Welfare Oblivion: Health and Mental Health in Juvenile Justice in Brazil, Colombia and Spain

**DOI:** 10.3390/ijerph20115989

**Published:** 2023-05-29

**Authors:** Ángela Carbonell, Sylvia Georgieva, José-Javier Navarro-Pérez, Mercedes Botija

**Affiliations:** 1Department of Social Work and Social Services, University of Valencia, 46022 Valencia, Spain; Angela.Carbonell@uv.es (Á.C.); Mercedes.Botija@uv.es (M.B.); 2Department of Developmental and Educational Psychology, University of Valencia, 46010 Valencia, Spain; Sylvia.Georgieva@uv.es

**Keywords:** mental health care, care systems, juvenile justice, Brazil, Colombia, Spain

## Abstract

(1) Background: This study aims to examine and describe the policies of three Latin American countries: Colombia, Brazil, and Spain, and identify how they implement their support systems for health, mental health, mental health for children and adolescents, and juvenile justice systems that support judicial measures with treatment and/or therapeutic approaches specialized in mental health. (2) Methods: Google Scholar, Medline, and Scopus databases were searched to identify and synthesize of the literature. (3) Results: Three shared categories were extracted to construct the defining features of public policies on mental health care in juvenile justice: (i.) models of health and mental health care, (ii.) community-based child and adolescent mental health care, and (iii.) mental health care and treatment in juvenile justice. (4) Conclusions: Juvenile justice in these three countries lacks a specialized system to deal with this problem, nor have procedures been designed to specifically address these situations within the framework of children’s rights.

## 1. Introduction

The general populations’ mental health is one of the most important pivot axes for the well-being of citizens. Mental health is considered a problem given its high prevalence and its detrimental consequences for the development of childhood and adolescence; thus, it is a concern and focus of future policies [1]. A very significant number of psychiatric disorders have their origin or turning point during adolescence [2]. According to the World Health Organization [3], a percentage close to 50% starts at the age of 14 years and 70% before the age of adulthood. In those diagnosed with mental health problems, a very high percentage occurs during childhood and adolescence, and what is more worrying, according to Escobar-Córdoba and Restrepo-Gualteros [4], a percentage much higher than the average is not attended by professional resources specialized in mental health, and much less in child and adolescent mental health.

Isolation, stigma, and labels are different stressors associated with children and adolescents diagnosed with a mental health pathology. Many obstacles that accompany their socialization process are directly related to their diagnosis [4]. This fact increases their self-enclosure and therefore the loss of prosocial relationships leading to a break with their healthy socialization process. This generates difficulties in the relationships with their neighbors, in their neighborhood, and with emotion management among others. These disconnections may encourage deviant behaviors or behaviors opposite to social control mechanisms [5].

Young people engaging in antisocial behavior are subject to social control measures, resulting in their placement in various juvenile justice systems separate from the adult criminal justice system. According to a study by Carl et al. [6], the highest percentage of mental illness among youth in institutional care is in residential settings. Another study points out the high diagnostic recidivism in young offenders, and that lack of control, impulsivity, and emotional instability act as facilitators for the development of antisocial behaviors [7]. In addition, drug use and abuse are additional problems presented by this population of young people that, along with mental disorders, generate co-morbidities with complex evolutions and treatments. In a study conducted by the Colombian Institute of Family Welfare [8], it was found that 83.9% of young people in the criminal justice system had used psychoactive substances, while only 17.3% had received treatment. This indicates that accessibility to treatment and the presence of barriers are particularly challenging issues.

Juvenile justice systems often prioritize the restitution of harm caused by the offender and the re-education of the young person rather than addressing the underlying factors that contribute to their criminal behavior. The absence of thorough diagnoses not only affects the behavior of young people but also hampers their access to treatment and gives unequal assurances of it. In this context, mental health problems become chronic.

Models of health and mental health care for any group are a mirror of political decisions aimed to increase citizens’ welfare. Responses to the needs of at-risk groups with mental health problems are not a policy priority as resources are often scarce, assistance-based and, as Dimenstein [9] notes in Brazil, fragmented and intermittent.

In the scientific literature, the study of mental health care for young offenders is also scarce, since research that delves into this topic is weak. There are barriers due not only to data protection of minors under judicial measures but also due to the stigma with which mental health is conceived. Consequently, the scientific literature has not been very prolific in this type of research given the complexity of its object of study [2].

De Almeida et al. [10] point out that Spain and Brazil are similar in terms of their administrative structure, guiding principles of health, psychiatric reform characterized by their deinstitutionalization plan, and a comprehensive funding system of universal coverage. However, they also highlight important challenges and gaps, such as the need for adequate access to mental health services for young people subjected to deprivation of liberty in the juvenile criminal justice system. Gathering the conclusions of a study conducted in Colombia by Castaño-Pulgarín and Betancur [7], it is considered necessary to build a concept of mental health in accordance with the characteristics and contexts of risk in which children and adolescents currently live. This study is carried out with the aim of examining and describing the policies of three Latin American countries, Colombia, Brazil, and Spain, and identifying how they implement their support systems for health, mental health, mental health for children and adolescents, and juvenile justice systems that support judicial measures with treatment and/or therapeutic approaches specialized in mental health.

## 2. Materials and Methods

A descriptive comparative study was carried out on mental health care policies aimed at young people under judicial measures for committing criminal acts in three different countries, Colombia, Brazil and Spain, in order to offer answers that contribute to the articulation of policies adjusted to the needs previously described. In the first stages of the research, we did not propose a comparative objective among them. However, the structured presentation of results by country, in order to adequately integrate the particular characteristics of each one, indirectly forced us to provide analogous and divergent views on the object of study.

This study was carried out through the screening and analysis of scientific publications, focusing on the creation, organization, financing, institutional public policies, management, resources, challenges, etc. of the models of health care and mental health care for children and adolescents in the countries under study.

### 2.1. Action Plan and Study Selection

The methodology was organized in a two-phase document search strategy. The first phase involved a review and analysis of the bibliography through an in-depth search of the scientific literature on the analysis of public policies on mental health care, child mental health, and mental health for adolescents in conflict with the law in Colombia, Brazil, and Spain. For this purpose, international databases such as Google Scholar, Medline, and Scopus were examined. The perimeter of borderline words to access scientific literature was configured according to Pati and Lorusso [11]: care policies, mental health, and juvenile criminal treatment. Therefore, three groups of keywords were identified and combined according to each section: (1) descriptive elements on mental health systems (e.g., “mental health policies”, “mental health network”, or “child and youth mental health”), (2) main concepts on justice (e.g., “juvenile justice”, “young offenders”, “judicial measures”, or “adolescents in conflict with the law”) and (3) descriptors such as “adolescents” or “adolescence” and “youth”. Additionally, the bibliographic references of each study were also analyzed in order to find other publications of impact.

Secondly, a comprehensive search for mental health and juvenile justice policy guidance documents was conducted using Internet search accelerators, institutional websites, and government administrations, including legal acts, laws, conferences, regulations, ordinances, protocols, informative regulations, and technical provisions. The search period was between 2000 and 2021 in Portuguese, English, and Spanish languages, excluding articles focusing on COVID-19 or derived physical health consequences.

The inclusion criteria were: (1) scientific articles, legal-assistance documents, and gray literature published between the years 2000 and 2021, (2) articles and working papers with the aforementioned keywords in English, Spanish, and Portuguese, (3) articles published on Google Scholar, Medline (PubMed), and Scopus databases, and (4) scientific articles that referred in any way to health care, mental health, childhood and adolescent mental health, and mental health treatment systems for adolescents in conflict with the law from three countries: Colombia, Brazil, and Spain.

### 2.2. Data Analysis

An inductive content analysis was conducted in line with the guidelines provided by Harper and Thompson [12]; the information was systematized by configuring areas and sub-areas based on text segments that were coded as emerging themes. Three conceptual categories were articulated to analyze mental health care policies aimed at adolescents in conflict with the law in Colombia, Brazil, and Spain: (1) health and mental health care models, (2) community-based child and adolescent mental health care, and (3) mental health care and treatment in juvenile justice.

## 3. Results

### 3.1. Identified Documents for the Revision

A total of 13,063 potentially eligible documents were identified for review. After eliminating duplicates, 11,081 were excluded for not meeting the inclusion criteria, including factors such as irrelevant subject matter, systematic reviews, meta-analyses, assessment instruments in young mental health, and recidivism in juvenile justice. This exclusion process resulted in 1982 articles, which were further assessed for relevance based on their title and abstract. As a result, a total of 33 articles were considered relevant and underwent a thorough reading of the full text. After this rigorous selection process, 24 articles were ultimately included in the qualitative synthesis, as depicted in Figure 1.

### 3.2. Health and Mental Health Care Models in Colombia, Brazil and Spain

The macro sphere is an intrinsic part of the policies that brings together trajectories, decision making, state pacts, consistency of agreements, international commitments, etc. This sphere facilitates the identification of decisions that affect socio-health regulations. It is also important to consider that there are nuances and differences in terms of population, economic development, political regime, and territorial and administrative structure (Table 1), among others. Colombia and Brazil have many similarities, despite geographic and language barriers and differences in size and population. The human development index and income per capita are similar, and this translates into comparable economies, spending levels, commodity prices, consumption habits, and social life. Picornell-Lucas and Pastor [13] state that Brazil, Colombia, and Spain present similarities both in their historical process as well as in the current organization of health and welfare models. Nevertheless, there are also differences focused on the management of health policies. This fact is detailed in a study by Delgado-Gallego et al. [14], in which it is explained that while Colombia stands out for its agile administration and longer consultation time, the case of Brazil converges with Spain in terms of greater availability of specialized personnel, materials, and high public spending and investment (especially the latter, in European countries). The cultural and historical characteristics of these countries are mainly related to the political changes and alteration of ideologies in power, i.e., neoliberal and progressive governments and a dictatorial legacy that in the case of Brazil and Spain was more disturbing and in Colombia more distant in time and with less repression towards the population, and less violence and stagnation of the welfare systems. These factors clearly triggered direct consequences for the orientation of health policies.

The redefinition of welfare systems and the regulation of new legal frameworks in the three countries contributed to the articulation of their health systems. In Colombia, the Health Care System was constituted in 1975 with the purpose of safeguarding the health of citizens with the help of instruments such as promotion, protection, and recovery through national, departmental, and municipal entities. The Colombian health care system has the characteristic of being formed by two coexisting systems: the contributory regime (private) and the subsidized regime (free) through a classification system based on the vital standard-Sisbén-. Both regimes provide universal coverage, equal access to medicines, surgical procedures, medical, and dental services. Through the Comprehensive Social Security System instituted by Law 100 of 1993, in its Title II, it was possible to bring together and coordinate a set of entities, rules, and procedures for universal coverage according to the principles of efficiency, universality, solidarity, comprehensiveness, unity, and participation [17]. Articles 186 and 227 establish the creation of a Quality Assurance and Accreditation System in Health. Subsequently, in 2015, the Statutory Health Law 1751 was approved, elevating it to a fundamental right, which was previously conceived as a mandatory public service, although with considerable restrictions. This article guarantees human dignity and equal opportunities for all Colombians in their right to receive health care. For its part, the Brazilian health system was built on the claims of the Health Reform Movement, which demanded from a Freirian perspective the participation of the subjects of the communities as agents of social transformation, therefore increasing social demand [18]. Finally, the Spanish National Health System (SNS) has its origin in 1986 and is free for citizens and residents affiliated to the social security system. This model converges with the Colombian General Social Security and Health System (SGSSS) as it relies on a system of Social Security contributions. Brazil’s Unified Health System (SUS) shares with the previous ones its public and universal nature, with common principles based on equity of access, decentralization, and community participation [19]. Despite these analogies, De Almeida et al. [10] argue that there are inequalities in access and quality among health services in these countries. In Brazil and Colombia, the private sector complements the assistance of public services through the Supplementary Health Care System (SAMS) and the quotas of the Colombian SGSSS [20]. Although the percentage of private health care affiliation in Spain has increased in recent years, it still falls short of one-fourth of the total population, with the majority opting for universal health care [21]. According to the framework laws, Colombia presents three levels of public health care: basic, specialized, and subspecialized, while private care only requires provision in two of them. Therefore, according to Prada-Rios et al. [22], there is an unbalanced coexistence of care. Specifically, Brazil and Spain have two levels of care, basic and specialized: (1) primary care or family health, located in outpatient units and health centers, and (2) specialized care, which includes emergency care, hospitalization, and rehabilitation. In both countries mental health care is integrated into the specialized level.

Regarding financing, the Colombian Public Health System is administered by the National Health Fund (FONASA), which operates on the basis of a collaborative scheme of quotas, financed by the State, active workers, and employer contributions [23]. In the case of Spain, the SNS is financed through general federal and autonomic taxes, and it is the autonomic governments that are responsible for the management and organization of health care in each territory [24]. The Sistema Unico de Saúde (SUS) is financed by taxes and social contributions from the three levels of government: federal, state, and municipal. SUS governance in Brazil is inter-federative and places the Ministry of Health as the coordinator and manager of the system while the states and the municipalities are the main executors of health policy and resource managers [25]. Currently, Colombia spends 2.9% of its GDP on public health and Brazil has 3.29%, compared to the 6.24% of spending in Spain (World Health Organization 2021). However, in the case of Mental Health, these numbers range between 3.3% in Spain (Ministry of Health, Consumption and Social Welfare 2020) and 1.4% and 2.3% in Colombia and Brazil, respectively [26]. On the other hand, Sweden and Finland, for instance, invest the highest percentage of their GDP in mental health, with numbers around 12% and 18%, respectively [27]; this fact suggests that Ibero-American countries still have much room for improvement [28].

Psychiatric reform in Spain began to be implemented after the approval of the 1986 Health Law and became a priority in Spain and also in other European countries. The aim of this law was to promote mental health and to prevent mental illness by guaranteeing a coordinated public network of resources as part of the health system [29]. One of the most important aspects of the Spanish health reform process was the incorporation of psychiatry into the general health system and the inclusion of public hospital psychiatric units. Currently, in Spain, there is no state mental health law, and, therefore, care is administrated by the laws, service portfolios, and strategic plans of each autonomous community and by the Mental Health Strategy of the NHS, which has been in the process of being updated since 2013 [30].

The strong demands and efforts of Brazilian mutualist entities, (i.e., collectives grouped in sick people, users, family members, etc.) allowed the beginning and subsequent vindication of changes towards a community model as an alternative to the psychiatric hospital [31]. As Moreira and Bosi [32] point out, psychiatric reform in Brazil intensified in the 1980s, but was not consolidated until 2001 with the enactment of Law No. 10,216. This law focused on the protection of the rights of people with mental illness and proposed a community mental health model. These actions led to the creation of the Psychosocial Care Network (RAPS) in 2011 by Ordinance No. 3.088/01 within the SUS.

Mental health policy in Colombia has been characterized by four important facts. The first is the integration of mental health in the field of public health since 1975. The second is the attempt to create a first national mental health plan that was aborted in 1998. The third, in 2013, was the attempt to implement the first National Mental Health Law promoted by the collaborative effort of the social and health sectors. Later, this law attempted to dissociate itself from positivism and the sick person conceived as an “object of” by including the social determinants of health, until, finally, in 2018, the National Mental Health Policy was created, and new hope emerged with the possibility of patient self-determination, placing the disease at the center of the map. This National Mental Health Policy assumed the positioning of mental health as a priority in the agenda for the country [33]. This policy was based on human rights, life course, gender, and population differential—territorial and psychosocial—within the model of social determinants of health and having an objective to promote mental health as an integral element of the right to health of all individuals, families, and communities. In other words, this law sought to encourage mental health and the mechanisms that make its development possible. Therefore, there are four important milestones in almost 50 years that have allowed the integration of mental health within the Colombian public health system.

The level of organization of the Colombian system is similar to the previous ones, as the gateway to mental health is the primary care system. In fact, 75% of mental health hospital beds are in public health management [34]. Coordination difficulties generate conflicts that stagnate or delay care. Hence, there are many integrated barriers in the system as it could have limited care resources available, since the use of mental health care services in Colombia does not cover the totality of people with mental disorders [33]. Zamora-Rondón et al. [35] conducted a study on the mental health system. The study identified several issues, including administrative, regulatory, and geographical gaps, as well as coordination problems and a low supply of health services. Additionally, the cultural context and social determinants were characterized by inequity and poverty, fragmentation of services, and few integrated networks. The primary care level was weak, and services were disjointed. Stigma persisted, human talent was insufficient, and there were few shared tasks. Institutional weakness prevented the enforcement of rights. In summary, the mental health system faced structural difficulties and fragmentation due to these issues [36].

The Brazilian network provides psychosocial care through a variety of facilities, such as Basic Health Units, Family Health Support Centers, Street Offices, Coexistence Centers, Reception Units, and residential care services. However, mental health care is mainly provided by Psychosocial Care Centers for adults (CAPS), children and adolescents (CAPSi), or individuals with disorders related to substance use and dependence (CAPSad). These centers offer services such as reception, assistance, psychosocial rehabilitation actions, and coordination [37]. In Spain, primary health care is the gateway to the mental health system and acts as responsible for early detection and clinical management for most mental health cases, as well as the referral to Mental Health Units (USM), Child and Adolescent Mental Health (USMIJ), Addictive Behavior Units (UCA), day hospitals, and rehabilitation centers, among other resources [38].

Unlike Brazil, Spain has a variety of residential resources with specific characteristics: supervised or supported housing, family respite care, therapeutic residences, etc. [10]. Spain has temporary hospitalization units for acute phases of the disease, interconsultations, day hospitals, therapeutic communities, and on-call and emergency services that operate in coordination with general hospitals. Dimenstein [10] conducted a study in Brazil and concluded that the implementation of comprehensive mental health care beds for crisis situations (general hospitals, CAPS III, emergencies, or substance abuse) is still a challenge. In the same line, Ardón-Centeno and Cubillos-Novella [39] state that in addition to the already described barriers, there are difficulties in implementing Mental Health Care Policies in Colombia based on the real needs of users. This occurs because the administrators/intermediaries of health resources, called EPS or Benefit Plan Management Entities (EAPB), which are both national and local, hinder the existence of community care networks. This barrier is due to the existence of a contracting system that is usually focused on economic profitability and not on the provision of services in the patient’s community.

There seems to be less distance between Spain and Brazil than between these two countries and Colombia, where the gaps in the public system itself affect the access to mental health. The first two have a greater diversification of resources and multidisciplinary teams for mental health care. However, according to Yoshiura et al. [40], the mental health service network in the three countries has developed rapidly. Despite this progress, the implementation of neoliberal socioeconomic policies has perpetuated the biomedical model and underfunded mental health policies. This has translated into limitations to meet the real needs of the population [41,42,43]. In 2017, Spain had 3.6 psychiatric beds per 100,000 inhabitants in general hospitals [44], while in Colombia these beds were 1.8 in the same type of hospital [28]. In the case of Brazil, there are currently 1622 beds for the entire country [45], which is less than one bed per 100,000 inhabitants. Trapé et al. [46] highlighted that in Spain, investment in mental health is equal for community care and hospital care, but in Brazil, investment in community care is 72%. Despite the fact that Brazil apparently invests more in community mental health and has more rehabilitation resources, both countries have psychosocial mental health care needs. Although these numbers have grown in Colombia in recent years, they are lower than in Brazil and Spain due to the aforementioned gaps, the distance between levels of health care, and the segmentation in insurance regimes that fosters inequalities [20]. Furthermore, the COVID-19 pandemic has required greater efforts in all lines, especially psychiatry [47]. Table 2 shows the aspects that characterize each health and mental health system.

### 3.3. Mental Health Care of Children and Adolescents

In 1977, the World Health Organization [48] established the recommendations to focus on the development of public systems of care for children’s mental health. This specialized health care developed belatedly. In fact, there were powerful claims at the end of the last century and the first decade of the 2000s with scientific studies and institutional reports denouncing the global absence of public interest in child and adolescent care [49,50,51,52]. Currently, only a small portion of mental health resources worldwide are allocated for child and adolescent care [53,54].

In Colombia, Góméz-Restrepo et al. [55] reported in the National Survey of Mental Health (2015) that 12% of adolescents had problems predictably connected to some mental pathology. Moreover, up to 70% of adults with mental illness report symptom onset in childhood. In this line, the activities of community psychiatric clinics have a notable lack of development, which directly affects the most vulnerable patients such as children and adolescents [56]. This is due to the scarcity of primary care resources and existing community devices. In fact, a study by Chaskel et al. [34] accentuated the lack of human resources working in Mental Health Care Services and Child Mental Health: “for the whole country about 900 psychiatrists (45 of them for children) and 1500 psychologists are responsible for providing mental health care in specialized medical centers, general hospitals and psychiatric centers”. In this sense, Law 1616 of 2013 was articulated with the objective of prioritizing mental health care for children and adolescents through health promotion and the prevention of mental disorders. In the words of Velásquez [57], Colombia has a broad normative framework on the protection of children’s rights, but the guarantee and execution of these rights is not so assuring. Thus, despite the fact that children’s mental health is a priority for the legal norms, the resources destined to cover these needs are very precarious and, therefore, they undermine their development.

According to Macedo et al. [58], in Brazil, the difficulty of structuring a specialized care system was related to the high complexity and variety of disorders involving children and adolescents: developmental disorders, externalizing disorders, suicidal and/or self-injurious behaviors, internalizing, or even substance abuse with early involvement of dual pathologies in the most severe cases. The enactment of the Brazilian Estatuto da Criança e do Adolescente (Law No. 8.069/90, ECA) in 1990, together with the psychiatric reform process and the implementation of mental health policies throughout the country, initiated a timid development of child and adolescent mental health care [59]. However, it was not until 2001, with the publication of Law No. 10.216/01 (stated in the institution of the Fórum Nacional de Saúde Mental Infanto-Juvenil and the holding of the III National Conference on Mental Health) that mental health for children and adolescents became a priority focus in psychiatric reform. It was then that policies began to be designed and implemented respecting the guidelines of the Brazilian deinstitutionalization process and the principles of ECA [60,61].

Free public psychiatric care for children and adolescents in Spain was absent until the 20th century. According to López-Fraile and Herrera [62], the democratic transition, the decline in the birth rate, and the emergence of new pathologies led to the prioritization of mental health for children and adolescents. Currently, most of the autonomous communities have set up a network of specific resources—outpatient units, day hospitals, or care in crisis situations—allowing them to offer quality care within a psychosocial approach. Despite these improvements, child and adolescent mental health care in Spain has been characterized by the absence of homogeneous criteria in their care model: no criteria to determine the maximum age of care, heterogeneity of approaches and treatments, lack of specific training in child and adolescent psychiatry, and lack of specialization in the face of the vulnerabilities. All of these affect the socialization and development of children and adolescents [1]. Moreover, there is an absence of specific plans and a scarcity of educational–therapeutic resources, which has led to the mental health care system being configured as a hodgepodge within the Spanish public health system, with negative consequences for patients [63].

Considering their particularities, Colombia, Brazil, and Spain have included in their care networks specific resources aimed at children and adolescents. Currently, the mental health of children and adolescents in Colombia is approached from a double dimension. The Colombian Institute of Family Welfare (ICBF) develops a series of programs whose axis pivots around the social determinants that affect children’s mental health. These actions start from prevention and focus their attention on positive socio-cognitive development and emotional management. The first filter towards specialized child mental health services takes place in primary care. Law 1616 established specific conditions for the care of children and adolescents involving prevention and integrated care services, psychiatric emergencies, specialized day hospitals for children and adolescents, integrated community mental health centers, and patient and family support groups. However, despite the fact that the regulations address all these spatialized services in child and adolescent psychiatric care, a study by Escobar-Córdoba and Restrepo-Gualteros [4] shows that “subspecialist dosctors in child and adolescent psychiatry provide their services mainly in Bogotá, with almost no presence in the less developed departments farther away from the capital”.

Brazil is attended in primary care facilities such as Basic Health Units and Family Health Strategy, Psychosocial Care Centers for Children and Adolescents (CAPSi), outpatient clinics, and general hospitals. According to Delfini and Reis [64], the integration of mental health in primary care allows for less stigmatizing care that is closer to the community. However, primary care teams are not always able to handle situations and need more specialization. Therefore, CAPSi functions as the main strategy in the face of the complexity of mental conditions for Brazilian children and adolescents [65]. CAPSi acts as a biopsychosocial rehabilitation service, offering individual, group, and family therapy on a daily or weekly basis depending on the needs of each adolescent [66]. In the case of Spain, the Child and Adolescent Mental Health Units (USMIJ) performs clinical treatment and outpatient follow-up of individual and family intervention [67].

### 3.4. Mental Health Care for Juvenile Offenders

The results of El Sayed et al. [68] showed that mental illness does not increase the risk of delinquent behaviors in minors, but that there are other more powerful risk factors such as drug abuse and psychosocial environments of risk or criminal versatility that determine recidivism. Even so, most of the studies on mental health and juvenile justice conducted in Colombia, Brazil, and Spain are focused on psychopathologies [69,70], new violence [71], neighborhoods, favelas and communes [72,73], criminal profiles [42], and risk of recidivism [74,75], among others. In Brazil, there are several studies on mental health care in young people with risk behaviors and drug use [76,77,78]. In the same line, Colombia has presented some studies in comparison with other countries that combine both dimensions: mental health care and models of care for adolescent offenders [79] and therapeutic assistance in the face of juvenile transgressions [80]. In Spain, no studies have been reported that jointly address the mental health of children and juvenile criminal justice systems. However, a recently published article partially addresses this dimension, focusing on the analysis of psychological care as an entry filter to mental health treatment [81]. The study included a sample of children and adolescents institutionalized in both the protection and juvenile criminal justice systems. Their results showed that 54.2% of juvenile offenders institutionalized in juvenile justice centers received psychological treatment.

Under Spanish law, a minor offender is a person between fourteen and eighteen years of age who commits acts defined as crimes or misdemeanors in the Spanish Criminal Code or special criminal laws, and who enters the juvenile justice system through Organic Law 5/2000, of 12 January, regulating the criminal responsibility of minors (LORPM). Among the different judicial measures, the LORPM regulates the following topics: reprimands, weekend stays, probation, day centers, cohabitation with other persons, family or educational groups, community services, outpatient treatments, or internments in different regimes (closed, semi-open, open, and therapeutic) associating different degrees from semi-liberty to absolute confinement.

Mental health care for adolescents in conflict with the law is carried out in two areas. First, outpatient mental health treatment aims to promote social reintegration and compliance with the judicial measure, ensuring emotional stability and balance through individual psychotherapeutic care. This care is mainly carried out from primary care devices or USMIJ [82]. Secondly, the LORPM establishes among its measures the so-called “therapeutic internment”. This measure is carried out in residential centers that provide specialized educational care or specific treatment aimed at minors with mental health or substance abuse problems. Boscà-Cotovad [83] defines these centers as complex institutions located at the intersection between infant and juvenile mental health and the juvenile penal system, where a problem is addressed by attending to clinical, judicial, family, academic, and social factors of the minor. This care is carried out from a multidisciplinary therapeutic perspective (psychological, psychiatric, and social), in coordination with a person in charge of the UMIJ of the area, and intervening in the different environments in which the minor interacts. Carbonell et al. [84] argue that the public network of mental health and addictive behaviors should provide coverage to juvenile justice centers. Specifically, by providing professionals with the necessary interventions and including counseling services and an adequate therapeutic offer for adolescents and young people in the open regime who require it. However, therapeutic internment should be considered as a strictly health management service.

In Colombia, the System of Criminal Responsibility for Adolescents (SRPA) is focused on the restoration of a deviant act, rather than on sanctioning or punishing the offender. This is carried out through protective, pedagogical, and restorative procedures and sanctions whose main objective is to restore the rights and social inclusion of the offender [85]. In 2006, Law 1098 was enacted, under the name of the Code of Childhood and Adolescence. This law aims to guarantee children and adolescents a positive development, advocating for the involvement of the victims and the family environment of the prosecuted individual.

The system distances itself from punitive justice and understands crime as a social conflict causing damage that can be restored. The system identifies minors under 18 years of age as subjects responsible for damage caused and also responsible for repairing it [86]. The sanctions imposed by Law 1098 are admonishment; imposition of rules of conduct; community service; assisted or supervised liberty; semi-closed sanction, which, depending on family support, can be more or less rigorous; and deprivation of liberty. Children and adolescents between 14 and 18 years of age with psychic or mental disabilities are considered criminally imputable. In order to improve rehabilitation, other types of assistance measures that affect mental health treatment were added, such as internment in drug rehabilitation centers. However, in the Colombian system, there is no measure that includes therapeutic actions as such [87].

In Brazil, the Statute of the Child and Adolescent (ECA), instituted by Law No. 8.069 of 13 July 1990, considers a child to be a person up to twelve years of age, and an adolescent to be a person between twelve and eighteen years of age. When they commit an infraction, adolescents are subject to a special process of accountability. The judge may apply measures in a regime of liberty or consider compliance with socio-educational measures, which may be extended up to 21 years of age. These measures occur in centers managed by the state governments in a regime of internment or semi-liberty. The Comprehensive Health Care Policy for Adolescents in Conflict with the Law in a regime of internment and provisional internment (PNAISARI), approved in Ordinance No. 1426/2004, details the operationalization of the policy. This policy provides specifications on funding, federative responsibilities, the organization of socioeducational and health services, and tools to manage intersectoral work [88]. In this line, socioeducational measures, regulated by Law No. 12,594/2012 that institutes the National System of Socioeducational Assistance (SINASE), present a gradation depending on the adolescents’ ability to comply with them, the circumstances, and the seriousness of the offense: warning, obligation to repair the damage, provision of services to the community, assisted liberty, insertion in a semi-open regime, and admission to socioeducational units [89]. A study by Costa and Silva [77] suggests that these socioeducational measures are based on authoritarianism and an absence of pedagogical activity by socioeducational agents.

Through Ordinance No. 1082/2014, PNAISARI has been redefined to ensure and expand access to health care for adolescents in conflict with the law within SUS, encompassing health promotion, prevention, assistance, and recovery, including mental health care [90]. The request for medical, psychological, or psychiatric treatment for adolescents in conflict with the law is regulated in Article 101 of the ECA, under inpatient or outpatient treatment.

Although the structure and regulation of mental health care in juvenile justice seems to be standardized, all three countries face major challenges when it comes to the mental health care of this group. Thus, mental health in juvenile criminal justice has traditionally been precarious. Navarro-Pérez et al. [91] point out that, despite the increase in pathologies, more and better health resources have not been provided to satisfy the needs of this group in Spain, nor have mixed socio-health care programs been integrated, that is, among the public health, judicial, and social administrations. On the other hand, Fernandes et al. [92] and Ribas and Canalias [93] highlight that health professionals in public services generally invisibilize adolescents, and this stigma is much greater in adolescents in conflict with the law, especially those deprived of liberty or with chemical dependencies.

In Brazil, the Ministério da Saúde [94] conducted a study on mental health care for adolescents deprived of liberty and its articulation with socio-educational units. The study found that in most cases, the mental health of adolescents was attended by internal services rather than psychosocial care services or SUS primary care. The study also revealed that institutionalization actions, such as containment, isolation, hospitalization, and medicalization, were frequently used. In Spain, Alcázar-Córcoles et al. [5] highlighted that most adolescents in conflict with the law do not arrive diagnosed to the juvenile justice system and do not adequately benefit from mental health care. This suggests a failure of the system as a whole (mental health, social services, and juvenile justice). Regarding Colombia, Law 1098 itself makes mental disorder invisible, labeling the subjects who require specialized therapeutic care as a “vulnerable population” without delimiting the specialized resources needed to implement psychiatric recovery and therapeutic treatments. In this line, the conclusions of the study by Arango-Dávila et al. [95] emphasized the urgent intervention of the Colombian State due to the epidemic nature of mental pathology.

All three countries are facing a common challenge, which is the pathologization of juvenile justice. As a result, there is an increase in medicalization and chemical containment used as a form of control over adolescents who comply with judicial measures. Attention is focused on repairing the harm caused, rather than addressing the psycho-social circumstances that led to it. Although legal systems hold up the concept of “reparation” as a banner, it is nothing more than a chimera far from reality. Velásquez [57] argues that the problem is not the lack or precariousness of normative regulations. The results of Scisleski et al. [96] highlighted that most hospitalizations were court-ordered and motivated by diagnoses that did not require invasive treatments. These diagnoses included conduct disorders due to the use of psychoactive substances or emotional and behavioral disorders. According to these authors, the use of court-ordered psychiatric hospitalization for adolescents with conduct disorders could be a new way of managing child and youth poverty, with the aim of perpetuating segregation. Costa and Silva [77] point out that the extreme medicalization of these adolescents may hinder the detection of more severe disorders. Delving into this line of analysis, Massó [97] additionally notes that believing that drug withdrawal programs will solve the mental health problems of juvenile offenders is not addressing psychiatric pathologies in their most complex dimension. Mental health in juvenile justice cannot be reduced only to the treatment of drug addiction.

## 4. Discussion

Authors should discuss the results and how they can be interpreted from the perspective of previous studies and of the working hypotheses. The findings and their implications should be discussed in the broadest context possible. Future research directions may also be highlighted.

The objective of this study was to investigate and describe the policies of three Latin American countries (Colombia, Brazil, and Spain) regarding the implementation of their healthcare systems, specifically in the areas of mental health, mental health services for children and adolescents, and the juvenile justice system’s utilization of specialized mental health treatment and therapeutic approaches within judicial measures. The analysis was carried out from the general to the particular, first analyzing the health care of each state to descend to mental health care, child and adolescent mental health, and, specifically, that which is articulated from the juvenile justice systems. Therefore, a broad approach to examine the object from its globality in achieved. The health and mental well-being of children and adolescents, along with the associated mental health issues, have a significant impact on the current and future generations. Hence, it is crucial to establish effective coordination among public welfare systems to ensure comprehensive socio-health interventions. The findings of the study support this notion, demonstrating a shared concern to establish evidence-based practices that foster coordinated and synergistic actions across health, mental health, and the juvenile justice system. However, the study also reveals that this interest is often undermined by historical factors, political inconsistencies, resource allocation, budget limitations, and specialization, as indicated by the results.

Historical processes have influenced the development of community models, which included mental health in the entry filter to the system through primary health care. Spain has a higher level of mental health development than Brazil and Colombia in this order. Nevertheless, the child and adolescent mental health systems of the three countries are still in the process of consolidation, especially for Brazil and Spain, and perhaps of growth for Colombia. Despite the prevalence and complexity of mental illness in adolescents in conflict with the law, this study shows the scarce scientific interest in mental health care policies aimed at this group in all three countries.

The few academic studies and government reports that analyze mental health care in adolescents in conflict with the law show a myriad of challenges related to stigma, under-diagnosis, complexity of disorders, lack of coordination between social, health, and judicial systems, lack of professional competencies, lack of specialization, scarcity of resources, etc. The results of Carbonell et al. [84], address biases towards mental illness and juvenile delinquency, highlighting, especially, the absence of generalist public resources for mental health care that should be claimed as universal.

In that sense, numerous studies from around the world on adolescents at risk point out that institutionalization policies are less effective than community-focused interventions [92], as in many cases institutional dependencies are generated [98,99]. The models established in Brazil and Spain tend towards the institutionalization and consequent stigmatization and do not opt for other alternatives of socio-educational care in the community. Colombia, for its part, must still consider which model of mental health will be promoted. On the one hand, there is a model that is institutionally regulated, which, according to Rojas-Bernal et al. [100], is subject to constant criticism and operational problems. On the other, maybe finding an alternative model that manages to tackle the real needs of Colombians, starting by multiplying resources and specialized professionals from public investment in mental health care for adolescents and youth at risk. The long internal armed conflict and the frequent situations of complex emergencies and disasters that have affected the country in recent years [101] set the country at the crossroads of betting on a comprehensive model.

Mental health in juvenile justice is a black hole as it is the disaster box of disruptive behaviors that occur in situations of social maladjustment [102], from behavioral treatments to cognitive therapies, to sexual aggressors, psychiatric disorders, psychotic outbreaks, problems deriving from drug abuse, and any other situation that has no a priori known or recognizably observable response. Faced with this flow of conflicts, a univocal system that regulates this type of event from an integrated dimension and that affects socialization is absolutely necessary. Some juvenile justice systems are light years away from a complete treatment in Mental Health; this is the case with Colombia. According to Massó [97], the approach to drug dependence is the tip of the mental health iceberg in juvenile justice.

Juvenile justice models have attempted to move away from penitentiary ones, with their own regulations away from adult criminal justice [88]. However, it has not been the same for the case of mental health. Mental health care for adolescents in conflict with the law should not be bound to the content of their judicial measure. Instead, the principle of resocialization and the best interests of the minor should prevail as guarantees of care and intervention. Thus, diagnosis, care, and treatment should be derived from public and community resources of specialized mental health. Therefore, the public network should be an opportunity to meet the needs of adolescents serving juvenile justice sentences. It should provide support, with the backing of other welfare administrations––justice and social services––regulating quality standards in patient care.

Enhancing public health systems to provide specialized mental health services to children and adolescents should be considered a top priority. The results of this study show the need to establish a system that prioritizes care for at-risk groups, such as adolescents in conflict with the law, and consider not only pathologies but also psychosocial treatment by specialized professionals to improve quality of life, increase awareness of illness, and reduce the risks associated with risky activities. These actions could be generated through a mixed model of specialized care, based on the public health system and with the collaboration of justice and community social services [103].

## 5. Conclusions

The research findings draw attention to the significant role played by the healthcare system in addressing mental health issues. Despite its crucial importance in maintaining population well-being and quality of life, public health systems often lack the necessary capacity to effectively support various aspects of mental health. Consequently, the vulnerability and fragility of the health system become apparent, particularly when other welfare systems, such as the justice system, rely on it to address mental health needs. Insufficient resources and inadequate responsiveness to the population’s mental health requirements further contribute to this challenge.

In this sense, the incomplete construction of health policies in Colombia, Brazil, and Spain has shown us diversified cultural, historical, and organizational aspects on the same background, where mental health appears as a residual concept within the approach to juvenile delinquency.

This study presents several limitations. Firstly, the search for articles was conducted in English, Portuguese, and Spanish, potentially excluding relevant studies conducted in other languages, particularly in countries with established welfare policies such as the Nordic countries. While English-language studies from these countries might have been included, it is also possible that relevant studies in Scandinavian languages were missed. Another limitation is the selection of journals, which were chosen from renowned databases but excluded the possibility of including results from the Clarivate WOS database to avoid duplications with other databases, such as Scopus, which has a larger number of indexed journals. Additionally, a limitation arises from the limited interest of scientific researchers in exploring and publishing articles on this particular topic due to difficulties in accessing information and navigating through permissions and authorizations. Furthermore, the scarcity of funded projects with objectives related to this complex field also contributes to the limitations. As mentioned earlier, there were articles that exceeded the scope of the study, focusing on more comprehensive topics such as caregivers or challenges within the juvenile justice system, as well as health risk practices unrelated to mental health treatment, such as infectious diseases or reproductive health. This unintentionally sidelines other areas of treatment that are not specifically integrated into the juvenile justice system, such as mental health treatment for adolescents in conflict with the law.

However, there are promising prospects on the horizon that encourage further exploration in this field of research. For instance, future studies could delve into the continuity of mental health treatment for adolescents who undergo probation measures followed by periods of confinement or social isolation. Additionally, investigating the characteristics of outpatient care for young offenders, the specific requirements for child psychiatrists specializing in juvenile justice compared to their counterparts, and the existing gaps and challenges in public mental health policies that affect young people in conflict with the law would contribute to a much-needed transformation.

Child and adolescent mental health, along with public policies for mental health care in juvenile offenders, require substantial changes. A socio-healthcare model should be established that diverges from the biomedical, institutional, and pharmacological approach that has long dominated adult psychiatry. The study also highlights that regulations often outpace the actual delivery of care, as the resources available fail to meet the regulatory requirements. Moreover, it emphasizes the necessity for the public health system to adequately respond to the needs of other welfare systems, offering both generic and specialized resources that facilitate the rehabilitation and recovery of individuals seeking support.

## Figures and Tables

**Figure 1 ijerph-20-05989-f001:**
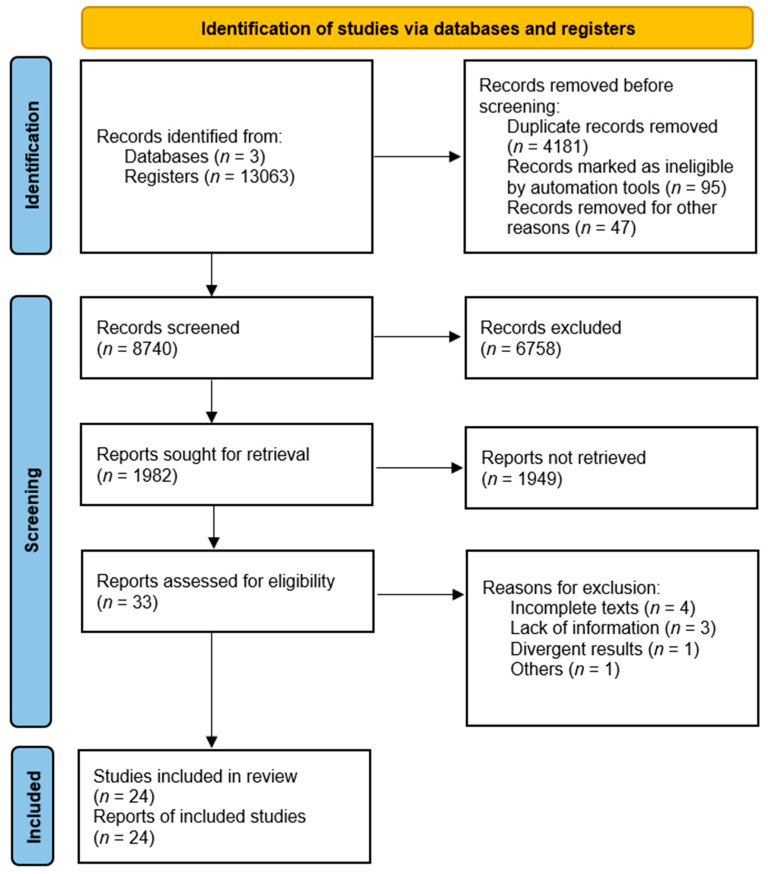
PRISMA 2020 flow diagram.

**Table 1 ijerph-20-05989-t001:** Socioeconomic Descriptors for Brazil, Colombia and Spain.

	Colombia	Brazil	Spain
Population	51.5 million	209.5 million	46.9 million
Current Government	Presidential Republic	Presidential federal republic	Parliamentary monarchy
Military dictatorship	1953–1957	1964–1985	1939–1975
GDP per capita	6.104 USD	7.507 USD	30.103 USD
HDI	0.752	0.761	0.893
HDI inequality	0.595	0.574	0.765
GDI	0.725	0.695	0.788

Notes: GDP = Gross Domestic Product; HDI = Human Development Index; HDI inequality = HDI adjusted for inequality; GDI = Gender Inequality Index. Source: Own elaboration based on World Bank [15] and United Nations [16].

**Table 2 ijerph-20-05989-t002:** Characteristics of health and mental health systems in Colombia, Brazil and Spain.

	Colombia	Brazil	Spain
Legal framework	Constitution of 1991 (arts. 44, 48, 49, 365 and 66)	Constitution of 1988 (arts. 196–200)	Constitution of 1978 (arts. 41 y 43)
System creation	Law 100 of 1993, Title II	Law 8080/1990 Unified Health System	Law 14/1986 General Health
Governance	Departmental and Capital District	Municipal interfederative	Autonomous Communities
System financing	Central Government. Departmental/District and Municipal	Federal, state, and municipal governments	Central and regional governments
Public expenditure on health	2.9% GDP per capita	3.29% GDP per capita	6.24% GDP per capita
Public expenditure on mental health	1.4% of health expenditure	2.3% of health expenditure	3.3% of health expenditure
Focus of care financing	60% hospital 40% community	28% hospital 72% community	51% hospital 49% community
Focus of attention	Primary Care	CAPS	Primary Care
Psychiatric beds in General Hospitals	1.1 beds/ 100,000 hab.	1.6 beds/ 100,000 hab.	3.6 beds/ 100,000 hab.

## Data Availability

The information used in the analysis is accessible from the scientific data sources.

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
