# Peer review of "From Social Rejection to Welfare Oblivion: Health and Mental Health in Juvenile Justice in Brazil, Colombia and Spain"

_ijerph, 2023, doi:10.3390/ijerph20115989_

Round 1

Reviewer 1 Report

Dear authors,

Here are some comments that I hope you will find useful for the publication of your manuscript.

1. Thank you for the conscientious and meticulous work, review of specialised literature on the subject under study.

2. The method section is very clear and has a very important function: to highlight the importance of pre-analyses.

3. As you deal with a triple social reality, the main doubt I have is related to the organisation of all the information you handle and that you present. 

In general, there is a tendency to talk about each country under each heading following a paragraph structure. However, I would like to share three ideas with you, in case you find them useful:

a) In the first heading this is a bit more confusing, as there are many implicit "subheadings" that inhabit it and sometimes the jump from country to country ends up confusing the reader who is forced to pay close attention   to not get lost in the geography of the mapa mundi. In this sense, I suggest that you review the first epigraph, and see if it is possible to regroup the "sub-epigraphs" that inhabit in a veiled way, so that there is not so much jumping from country to country, from theme to theme.

b) In general, I would opt for a redundant scheme in each epigraph, whereby a general conceptual introduction is made to the topic addressed by each epigraph, followed by the realities (balanced, documented and shared) of the countries, and closing with a conclusion to each epigraph that reflects the reality that has just been presented. That is to say, it is important for the reader that if, for example, Colombia has already been mentioned in the first paragraph, it is not mentioned again in the last paragraph as a conclusion. I think it is necessary for the introduction and closing of each heading to be general/global/without any specific country. 

c) In this same sense, the section on conclusions and discussion is very good. I don't know if it would be possible to include in a few lines the similarities and differences between these countries in terms of the topic under study.

In any case, it is a well-written and highly publishable manuscript, which lays a worthy foundation for future research on the subject.

Thank you for your effort and work.

Author Response

Please see the attachment. Our responses are written in blue color.

Reviewer 2 Report

Dear Editors, thanks for allowing me to review this exciting paper titled "From Social Rejection to welfare oblivion: Health and Mental Health in Juvenile Justice in Brazil, Colombia and Spain". The paper meets a very relevant issue and an unmet topic in literature. Particularly relevant is the comparison between countries, EU and not EU. Moreover, the paper is well-written and easy to read. 

Unfortunately, the paper shows some critical aspects requiring in-depth article revision. For this reason, I suggest a Major revision. I hope the authors do not feel demotivated by this result.

Abstract: must be revised after revising the other part better to emphasise the study’s aims and originality.

Introduction:

First thing: the aims must be better declared and clarified. Do you want to compare the policies to define the support systems existing in the studied three countries to identify differences between countries by country? I suppose this is only after reading the result, discussion, and conclusion.   Is this not your aim?, please, you have to declare it better.

 Moreover, you must better underline the originality of your work related to :

1- comparative study or a multi-country study, including EU and non-EU countries. Literature usually  proposes one country study on analysis  of the system or comparing  similar countries geographically  closer  ( e.g. Fagan, A. A., Bumbarger, B. K., Barth, R. P., Bradshaw, C. P., Cooper, B. R., Supplee, L. H., & Walker, D. K. (2019). Prevention Science, 20, 1147-1168.) scaling up evidence-based interventions in US public systems to prevent behavioral health problems: Challenges and opportunities.

2. there are some scoping reviews o mental  in literature,( e.g. Borschmann, R., Janca, E., Carter, A., Willoughby, M., Hughes, N., Snow, K., ... & Kinner, S. A. (2020). The health of adolescents in detention: a global scoping review. The Lancet Public Health, 5(2), e114-e126.)  Please better explain the contribution of this study why this is different?  for issue? methods ?  

3 . Please better underline how  recent literature  discusses:

The issue of mental health related to:

a) covid 19 pandemia effects ( e.g.  e.g. Gordon, F., Klose, H., & Lyttle Storrod, M. (2021). Youth (in) justice and the COVID-19 pandemic: Rethinking incarceration through a public health lens. Current Issues in Criminal Justice, 33(1), 27-46; Barnert, E. S. (2020). COVID-19 and youth impacted by juvenile and adult criminal justice systems. Pediatrics, 146(2).)

b)  focus the attention on different targets of people as well as older people and caregiving ( e.g. Riedel‐Heller, S. G., Busse, A., & Angermeyer, M. C. (2006). The state of mental health in old‐age across the ‘old’European Union–a systematic review. Acta Psychiatrica Scandinavica, 113(5), 388-401;

Gagliardi, C., Piccinini, F., Lamura, G., Casanova, G., Fabbietti, P., & Socci, M. (2022). The Burden of Caring for Dependent Older People and the Resultant Risk of Depression in Family Primary Caregivers in Italy. Sustainability, 14(6), 3375.)

 c)  focus the attention o specific aspects/  characteristics of mental health, as well as loneliness and depression issues ( e.g.Fox, D. J., & Hanes, D. (2023). Prevalence and correlates of unmet mental health services need in adolescents with a major depressive episode in 2019: an analysis of National Survey on Drug Use and Health Data. Journal of Adolescent Health, 72(2), 182-188.;Casanova, G., Abbondanza, S., Rolandi, E., Vaccaro, R., Pettinato, L., Colombo, M., & Guaita, A. (2021). New older users’ attitudes toward social networking sites and loneliness: the case of the oldest-old residents in a small Italian city. Social Media+ Society, 7(4), 20563051211052905 )

Moreover, the literature does not present many revisions of policies and even in this issue   generally focus on a specific topic and with a different target  ( e.g      Zajac, K., Sheidow, A. J., & Davis, M. (2015). Juvenile justice, mental health, and the transition to adulthood: A review of service system involvement and unmet needs in the US. Children and youth services review, 56, 139-148; Salido, M. F., Moreno-Castro, C., Belletti, F., Yghemonos, S., Ferrer, J. G., & Casanova, G. (2022). Innovating European Long-Term Care Policies through the Socio-Economic Support of Families: A Lesson from Practices. Sustainability, 14(7), 4097).

Methods :

The methods section is too short and low detailed.

 How many papers and policies have been selected for the study?

What are the criteria for inclusion? More,

 why do you include "scholars" as an engine of research? It is usually not included in the review, but I can deduct and understand the reason may be, but you must declare it, even why not include Pub med (usually as the first engine search used?)

 generally, I miss a diagram of the study's process and another one on the study's conceptual framework. Please help the reader to understand what you did and why!

 How have you selected the three categories of analysis, and why?

Discussion and conclusion. Please separate the discussion from the conclusion

In  the discussion, you can cut the line from 515 to 517

 The discussion must be implemented the argumentation related to the aim of the study and the policy's implication. Moreover,  please indicate the limitation of the study and future studies 

The conclusion section must be implanted too

Author Response

(The authors gave the same response as above.)

Round 2

Reviewer 2 Report

Dear Editors and Authors, Many thanks for giving me the opportunity to review this interesting paper again. I would like to give thanks even to the authors to have considering most of the inputs.  The paper is significantly improved, in methods,  discussion and conclusion sections. Unfortunately, the relevance for literature and for the international context does not find the same valorisation, from my point of view ( of course) ed it is a very pity because the potential of this paper's suggestions could go over the specific topic, to meet a larger vision of mental care system existing in the countries, or related topics,  if well discussed.  The authors decided to not enforce my suggestion on opening the vision.  From my point of view, this closing does not benefit the originality of the paper and the weight of it in the literature. However, for not closing the opportunity to be published, I recommend "minor revisions", leaving to other reviewers' opinions the decision to publish. 

If the paper will have another time of revision, I suggest to the author revise the introduction to better meet the issues already underlined in the first set of my revision.   
